# Modelling and Research on Intuitionistic Fuzzy Goal-Based Attack and Defence Game for Infrastructure Networks

**DOI:** 10.3390/e25111558

**Published:** 2023-11-18

**Authors:** Zhe Li, Jin Liu, Yibo Dong, Jiaqi Ren, Weili Li

**Affiliations:** Science and Technology on Information Systems Engineering Laboratory, National University of Defense Technology, Changsha 410073, China; lizhe@nudt.edu.cn (Z.L.); liujin229234@nudt.edu.cn (J.L.); dongyibo@nudt.edu.cn (Y.D.); jiaqiren@nudt.edu.cn (J.R.)

**Keywords:** infrastructure networks, attack and defence games, intuitionistic fuzzy set, network science, repeated games

## Abstract

Network attack and defence games are gradually becoming a new approach through which to study the protection of infrastructure networks such as power grids and transportation networks. Uncertainty factors, such as the subjective decision preferences of attackers and defenders, are not considered in existing attack and defence game studies for infrastructure networks. In this paper, we introduce, respectively, the attacker’s and defender’s expectation value, rejection value, and hesitation degree of the target, as well as construct an intuitionistic fuzzy goal-based attack and defence game model for infrastructure networks that are based on the maximum connectivity slice size, which is a network performance index. The intuitionistic fuzzy two-player, zero-sum game model is converted into a linear programming problem for solving, and the results are analysed to verify the applicability and feasibility of the model proposed in this paper. Furthermore, different situations, such as single-round games and multi-round repeated games, are also considered. The experimental results show that, when attacking the network, the attacker rarely attacks the nodes with higher importance in the network, but instead pays more attention to the nodes that are not prominent in the network neutrality and median; meanwhile, the defender is more inclined to protect the more important nodes in the network to ensure the normal performance of the network.

## 1. Introduction

With the continuous development of science and technology, people’s daily lives are more and more inseparable from the support of various infrastructure networks. Whether it is a larger network such as the power network and aviation network, or a smaller network such as the underground line network, the paralysis of any infrastructure network will bring about immeasurable social impact and economic property losses. However, security incidents on infrastructure networks are frequent. For example, on 16 September 2022, a fire broke out in the Changsha Telecom Building, which affected the voice function of some users’ mobile phones, as well as caused huge economic and property losses. While these infrastructure networks bring great convenience to people’s lives, they also face great security and defence challenges, and they are more likely to become important military targets in wartime. Research points out the ever-growing importance of network resilience against targeted attack and error [1,2]. Therefore, how to respond to the attacks of terrorist organisations on infrastructure and how to make use of existing resources for effective protection are the focus of attention for the security and defence departments of various countries [3].

From the current information, most of the research on attack and defence games in infrastructure networks is now set in the context of deterministic conditions [4,5,6], which rely on objective indicators in complex networks for modelling [7,8], such as node degree, network efficiency, etc. In practice, however, the situation is so intricate and confusing that it is difficult to model and predict, with certainty, the realistic conditions faced by both attackers and defenders. In addition, the huge amount of data information involved in the game, as well as the subjective decision preferences of both attackers and defenders, further increase the uncertainty of the game. Therefore, it is necessary to more comprehensively consider the ambiguities and uncertainties involved in real-world problems on the basis of the existing research on attack and defence games in infrastructure networks.

This paper introduces the membership degree and non-membership degree of intuitionistic fuzzy sets to quantitatively reflect the uncertainty of game objectives induced by the subjective decision-making preferences of players, based on the existing objective attack and defense game models. This article’s game model gives more information for players’ strategy choices by developing the expectation value, rejection value, and hesitation degree of the target, bringing the game model closer to reality and making decisions more sensible. When compared to existing models, the game model in this article gives more information for players’ strategy choosing, bringing the game model closer to reality and making judgments more rational.Therefore, the main work and innovations of this paper are as follows:Modelling offensive and defensive games from a network science perspective.An intuitionistic fuzzy, goal-based game model for the attack and defence scenarios of infrastructure networks is developed.Algorithms are designed to solve the developed model, and the results are analysed.The repeated game process is studied.

The rest of the paper is organised as follows: Section 2 contains the literature review of this study; Section 3 provides a detailed portrayal of the game model; Section 4 gives the methodology for solving the game model; Section 5 is the experimental analysis part of the paper; and Section 6 concludes the full paper.

## 2. Literature Review

Realistic critical infrastructure networks often have thousands of nodes, with individual nodes interconnecting and interacting with each other to form a complex network structure. In the late 1950s, the Hungarian mathematicians Erdős and Rényi proposed the stochastic network model [9], which initiated the study of complex networks. It was not until the recent last few decades—after Watts and Strogatz proposed the small-world network model [10] and Barabási and Albert proposed the scale-free network model [11,12], that the number of studies on complex networks increased significantly. Numerous scholars have studied complex networks from the perspective of complex network models, as well as analysed the topological features of different models and their applications [13,14,15,16], which have continuously promoted the development of complex network-related research.

With the increasing threats to infrastructure networks, game theory has become a powerful research tool in infrastructure network security problems. Furthermore, many scholars have combined game theory with critical infrastructure networks, as well as constructed different complex network attack and defence game models to study the attack and protection problems of critical infrastructure networks. Borwn et al. [17] applied game theory to military strikes and homeland defence as early as 2003, and they also conducted a great deal of studies based on it. In addition, they studied a dynamic game model through two-layer and three-layer planning models, as well as analysed optimal attack strategies under the three defensive strategies of no defensive behaviour, protecting critical nodes, and protecting three-quarters of nodes [18]. Li et al. [19,20,21] established a two-player static game model on complex networks, as well as studied equilibrium strategies and node degrees. The relationship between equilibrium strategies and node degree values, as well as the effects of network structure, cost constraints, and cost sensitivity on the equilibrium outcome were investigated. Fu et al. [22] added a dynamic game model in which the defender first protects the network through protection or camouflage behaviours, whereby the attacker then takes measures to destroy the network. Following this, they investigated the effect of the defender’s willingness on the defender by using evolutionary games. Gu et al. [23] developed a Bayesian Starkelberg game model to analyse the effect of type distribution on an equilibrium solution in the face of the attacker having different utility functions. Jiang et al. [24] developed a Bayesian Starkelberg game model to study the water supply network protection problem, including four private information situations. Zeng et al. [25,26] focused on the problem of scheduling critical infrastructure network defence resources under conditions of asymmetric information, thereby introducing the Stackelberg active deception game and the Bayesian Stackelberg game to build the model. Zhang et al. [8] investigated the optimal allocation of resources between the defender and the attacker’s multiple true/false objectives by building an asynchronous action game. Thompson et al. [27,28] developed and solved a defender–attacker–defender optimisation model to plan the optimal defence and response to intelligent attacks on the US air transport network. Yuan et al. [29] proposed to extend the traditional defender–attacker–defender model to post-incident corrective line switching operations as an effective post-incident mitigation method through which to protect the power infrastructure network. Abidi, M.H. et al. [30] proposed a vulnerability model for a CPS resource allocation process using a two-stage min-max game to minimise the value of loss costs. R. B. Benisha et al. [31] developed a two-step tracking model for UVI attacks against power grids using game theory ideas to improve detection efficiency and security. Li et al. [32] constructed an infrastructure network protection strategy solution model with “uncertain attack resources and limited protection resources”, as well as explored the effectiveness and practicality of the PSDA algorithm under conditions of incomplete information. Wang et al. [33] proposed an active defence strategy selection method for military information networks in the form of pure strategies. Jiang et al. [34] established an attack–defence two-layer game model, as well as a defence–attack–defence three-layer sequential game model for different infrastructure network attack–defence scenarios; following this, it was found that the effectiveness of the obtained defence strategies was significantly improved. As shown in Table 1, the literature review table provides a more visual display of related studies.

However, there are a large number of uncertainties in real situations that are difficult to model and to quantify effectively [35]. Moreover, the theory of intuitionistic fuzzy games provides a good idea for the solution of the above problems. In 1986, Atanassov, a Bulgarian scholar, proposed the concept of the intuitionistic fuzzy set (IFS) [36], which laid the theoretical foundation for the development of intuitionistic fuzzy games. Subsequently, Delgado defined two characteristics of fuzzy numbers [37], Angelov studied the intuitionistic fuzzy optimisation model [38], and Aggarwal gave a two-player zero-sum game solution method with the goal of the intuitionistic fuzzy set (which pushed the theory of the intuitionistic fuzzy game to maturity) [39]. In recent years, the establishment of the model in this paper is mainly based on the intuitionistic fuzzy goal zero-sum game model of Li and Nan [40,41].

## 3. Model

A complete game model is mainly composed of model assumptions, strategy sets, and payoff functions, and these aspects are mainly modelled and introduced in this section.

### 3.1. Symbol Definitions

The definitions and explanations of the symbols used in this paper are presented in Definitions.

### 3.2. Model Assumptions

The game model in this paper contains only two players—namely the attacker, that is, the player $p_{1}$, and the defender, that is, the player $p_{2}$. Under limited resource constraints, the offensive and defensive sides conduct offensive and defensive games around certain nodes in a complex network. Among them, the goal of the attacker is to destroy the topology of the complex network to the greatest extent by attacking some nodes in the network, thereby reducing the performance of the complex network and making it unable to maximise its effectiveness; the goal of the defender is to make the complex network work properly by protecting certain nodes in a complex network. In view of the game model in this chapter, the following assumptions are made.

In this game model, only one attacker and one defender are involved. And both sides have complete information, that is, they both know the complete topology of the complex network, as well as understand all the potential strategies and benefits of each other’s strategies. Furthermore, the offensive and defensive sides are absolutely rational, and they can respond accordingly according to each other’s actions to maximise their own interests.The attack and defence are targeted at the nodes. Attackers will choose certain nodes to attack, and defenders will choose certain nodes to protect. If a node is successfully attacked, the edges connected to the node will also be removed.If a node is attacked by an attacker without the protection of the defender, it will be considered to be successfully attacked and removed from the network. However, if the node is protected by the defender, the attacker’s attack on it will be deemed invalid, and the node will continue to play its role in the network.Attackers and defenders act simultaneously, that is, both sides do not know the specific strategies adopted by each other before they act. The game can be carried out in one round or multiple rounds. Specifically, the offensive and defensive sides deliver corresponding action strategies, as well as calculate the income values of the offensive and defensive sides under this strategy profile so as to obtain the income matrix. If a multi-round game is carried out, the game between the two sides will be carried out on a complex network after removing the successfully attacked nodes.The cost of an attacker attacking each node is the same and is not affected by the attributes of the node itself; similarly, the cost of defenders defending each node is the same.

### 3.3. Network Modeling

The complex networks studied in this game model are all unweighted and undirected graphs, which can be represented by G=(V,E), where $V=\left[v_{1},v_{2},\ldots ,v_{N}\right]$ represents the set of all nodes, and E⊆V×V is the set of all edges. Let N=|V| denote the number of nodes, then $A\left(G\right)={\left(a_{ij}\right)}_{N\times N}$ can be used to represent the adjacency matrix of the network. If there is an edge connection between the node $v_{i}$ and the node $v_{j}$, then $a_{ij}=a_{ji}=1$, otherwise, $a_{ij}=a_{ji}=0$, which means that there is no edge connection between the node $v_{i}$ and the node $v_{j}$.

### 3.4. Game Strategy

The network attack–defence game model studied in this paper aims at the nodes in the network; as such, the strategy set is the nodes in the network. It is assumed that the cost of both sides of the game to attack and defend each node is 1, the available resources of the attacker are $n_{A}$, and the available resources of the defender are $n_{D}$, that is, the total number of nodes that the attacker can attack at one time is $n_{A}$, and the total number of nodes that the defender can defend at one time is $n_{D}$. The vector $X_{A}=\left[x_{1},x_{2},\ldots x_{N}\right]\in S_{A}$ is defined as an attack strategy, where *N* is the number of nodes in a complex network, and $S_{A}$ is the set of all attack strategies. If the attacker chooses to attack the *i* node, then $x_{i}=1$, otherwise, $x_{i}=0$. From this we can obtain that, for any attack strategy $\sum_{i=1}^{N}x_{i}=n_{A}$ holds. Similarly, the vector $Y_{D}=\left[y_{1},y_{2},\cdots ,y_{N}\right]\in S_{D}$ represents a defensive strategy, where $S_{D}$ is a set of all defensive strategies, $y_{i}=1$ and $y_{i}=0$ represent the defender’s choice of defensive and non-defensive *i* nodes, respectively, and $\sum_{i=1}^{N}y_{i}=n_{D}$ exists for any defensive strategy.

According to the assumption in the previous section, when the node $v_{i}$ is not protected by the defender and is attacked by the attacker, it will be considered as successfully attacked and removed from the network, that is, when $x_{i}=1$ and $y_{i}=0$, the node $v_{i}$ will be removed. As shown in Figure 1, this is a schematic diagram of the attack and defence process around the node in the attack and defence game for infrastructure networks.

After a round of a game, the set of all removed nodes is $\hat{V}\subseteq V$, the set of all remaining edges after removing nodes is $\hat{E}\subseteq \left(V-\hat{V}\right)\times \left(V-\hat{V}\right)$, and the network at this time is recorded as $\hat{G}=\left(V-\hat{V},\hat{E}\right)$. Thus, $\hat{V}$ can be obtained by the following equation:(1)$$\hat{V}=V^{A}-V^{A}\cap V^{D}$$
Among them, $V^{A}\subseteq V$ is the set of all the attack nodes; $V^{D}\subseteq V$ is the set of all the defensive nodes.

### 3.5. Payoff Function

The payoff function is used to calculate the payoff of the offensive and defensive sides under different strategic profiles. In this chapter, we define the payoff function of both sides based on the network performance evaluation function. The network performance evaluation function can reflect the overall topology of the network and reflect the impact of different node deletions on the performance of the entire network. Generally, the maximum connected slice size, the number of reachable node pairs, and the network efficiency are used to evaluate the network performance. Furthermore, $U^{A}\left(X_{A},Y_{D}\right)$ is defined as the attacker’s payoff function, where $X_{A}$ is the attack strategy selected by the attacker, and $Y_{D}$ is the defensive strategy selected by the defender. The calculation formula is as follows:(2)$$U^{A}\left(X_{A},Y_{D}\right)=\frac{\Gamma \left(G\right)-\Gamma \left(\hat{G}\right)}{\Gamma (G)}\in \left[0,1\right]$$

Similarly, $U^{D}\left(X,Y\right)$ is defined as the defender’s payoff function, which is computed as follows:(3)$$U^{D}\left(X_{A},Y_{D}\right)=\frac{\Gamma \left(\hat{G}\right)-\Gamma \left(G\right)}{\Gamma (G)}\in \left[-1,0\right]$$
Among them, Γ is the network performance evaluation function, which does not increase monotonously with the decrease in the number of network nodes. Thus, the complex network $\hat{G}$ after the nodes are removed is a subgraph of the original complex network *G*, which means $\Gamma \left(\hat{G}\right)\leq \Gamma \left(G\right)$. This attribute indicates that the network’s performance suffers when nodes are removed.

This chapter selects the maximum connected slice size as an indicator to evaluate the network performance. The maximum connected slice size in a complex network refers to the number of nodes contained in the greatest connected subgraph, which determines the network’s reachability and information transmission efficiency. If there are several disconnected subgraphs in the network, there is no information exchange between them, and the network’s goals cannot be met. As a result, the maximum connectivity slice size can be utilized to assess the network’s connection and information transmission efficiency. The following formula can be used to calculate the maximum connected slice size:(4)$$L_{max}=max_{i=1}^{n}\left|C_{i}\right|$$
Among them, *n* represents the total number of nodes in the complex network, $C_{i}$ represents the connected subgraph where node *i* is located, and $|C_{i}|$ indicates the number of nodes in the subgraph.

## 4. Methods

In order to better describe the uncertainty in the game process and to make the network attack and defence game model more in line with the actual situation, the game model used in this chapter is an intuitionistic fuzzy target, zero-sum game model, and the solution method will be introduced below. According to the research of Nan and Li on the intuitionistic fuzzy goal zero-sum game model [40], we define our intuitionistic fuzzy goal zero-sum game model as the following multivariate array:(5)$$\mathrm{IFG}=\left(X,Y,F,v_{a},p_{a},v_{r},p_{r},\omega_{a},q_{a},\omega_{r},q_{r}\right)$$
Here, *X* and *Y* are the mixed strategy spaces of player $p_{1}$ and player $p_{2}$, respectively, and this is denoted by the following:(6)$$X=\left\{x\in R^{m}|\sum_{i=1}^{m}x_{i}=1,x_{i}⩾0,i=1,2,\;L,m\right\}$$
and
(7)$$Y=\left\{y\in R^{n}|\sum_{j=1}^{n}y_{j}=1,y_{j}⩾0,j=1,2,\cdots ,n\right\}$$
where the payoff matrix for player $p_{1}$ is concisely expressed as the matrix $F={\left(b_{ij}\right)}_{m\times n}$, and the payoff matrix for player $p_{2}$ is $-F={\left(-b_{ij}\right)}_{m\times n}$ since the matrix game is zero-sum. Among them, $v_{a}$ is the expected value of the player $p_{1}$ to the target, and the error is $p_{a}\geq 0$; in addition, $v_{r}$ is the rejection value of the player $p_{1}$ to the target, and the error is $p_{r}\geq 0$. Similarly, $\omega_{a}$ and $\omega_{r}$ represent the expected value and rejection value of the player $p_{2}$ to the target, respectively, and $q_{a}\geq 0$ and $q_{r}\geq 0$ are the corresponding errors, respectively.

Let $D=\left\{x^{T}Fy|\left(x,y\right)\in X\times Y\right\}\subseteq R$ be an intuitionistic fuzzy set on *D*, where the payoff goal of $p_{1}$ is $A=\left\{\langle x^{T}Fy,\mu_{A}\left(x^{T}Fy\right),v_{A}\left(x^{T}Fy\right)\rangle |x^{T}Fy\in D\right\}$. Among these, we can express them as follows:(8)$$\begin{array}{c} \mu_{A}:D\to \left[0,1\right]\;\\ x^{T}Fy\in D\;\mu_{A}\left(x^{T}Fy\right)\in \left[0,1\right] \end{array}$$
and
(9)$$\begin{array}{c} v_{A}:D\to \left[0,1\right]\;\\ x^{T}Fy\in D\;v_{A}\left(x^{T}Fy\right)\in \left[0,1\right] \end{array}$$
and satisfy
(10)$$0⩽\mu_{A}\left(x^{T}Fy\right)+v_{A}\left(x^{T}Fy\right)⩽1$$

Similarly, for the player $p_{2}$, its goal on the payoff value is also an intuitionistic fuzzy set on *D*, which is in the form of $B=\left\{\langle x^{T}Fy,\mu_{B}\left(x^{T}Fy\right),v_{B}\left(x^{T}Fy\right)\rangle |x^{T}Fy\in D\right\}$. Among these, we can express them as follows:(11)$$\begin{array}{c} \mu_{B}:D\to \left[0,1\right]\;\\ x^{T}Fy\in D\;\mu_{B}\left(x^{T}Fy\right)\in \left[0,1\right] \end{array}$$
and
(12)$$\begin{array}{c} v_{B}:D\to \left[0,1\right]\;\\ x^{T}Fy\in D\;v_{B}\left(x^{T}Fy\right)\in \left[0,1\right] \end{array}$$
and satisfy
(13)$$0⩽\mu_{B}\left(x^{T}Fy\right)+v_{B}\left(x^{T}Fy\right)⩽1$$

Among them, the membership degree $\mu_{A}\left(x^{T}Fy\right)$ is the acceptance degree of the player $p_{1}$ to the payment value, and the non-membership degree $v_{A}\left(x^{T}Fy\right)$ is the rejection degree of the player $p_{1}$ to the payment value; similarly, the membership degree $\mu_{B}\left(x^{T}Fy\right)$ and the non-membership degree $v_{B}\left(x^{T}Fy\right)$ represent the degree of acceptance and rejection of the payoff value by the player $p_{2}$, respectively.

It is assumed that the membership function and non-membership function of the player $p_{1}$ are linear as follows:(14)$$\mu_{A}\left(x^{T}Fy\right)=\left\{\begin{array}{cc} 0 & \left(x^{T}Fy<v_{a}-p_{a}\right)\;\\ 1-\left(v_{a}-x^{T}Fy\right)/p_{a} & \left(v_{a}-p_{a}⩽x^{T}Fy<v_{a}\right)\;\\ 1 & \left(x^{T}Fy⩾v_{a}\right) \end{array}\right.$$
(15)$$v_{A}\left(x^{T}Fy\right)=\left\{\begin{array}{cc} 1 & \left(x^{T}Fy<v_{r}-p_{r}\right)\;\\ \left(v_{r}-x^{T}Fy\right)/p_{r} & \left(v_{r}-p_{r}⩽x^{T}Fy<v_{r}\right)\;\\ 0 & \left(x^{T}Fy⩾v_{r}\right) \end{array}\right.$$

A schematic diagram of the membership function and non-membership function of the player $p_{1}$ is shown in Figure 2.

From the function image, we can obtain that, when $x^{T}Fy\leq v_{r}-p_{r}$, the satisfaction degree of the player $p_{1}$ is 0, and the rejection degree is 1; when $v_{r}-p_{r}<x^{T}Fy\leq v_{a}-p_{a}$, then the satisfaction degree of the player $p_{1}$ is 0, but because the player $p_{1}$ has a certain degree of hesitation, it will not completely refuse at this time; when $v_{r}<x^{T}Fy\leq v_{a}$, then the degree of rejection of the player $p_{1}$ is reduced to 0, but because the player $p_{1}$ has a certain degree of hesitation, it will not be fully satisfied at this time; and when $x^{T}Fy>v_{a}$, then the satisfaction degree of the player $p_{1}$ is 1, and the rejection degree is 0. It can be seen that the intuitionistic fuzzy set reflects the degree of satisfaction, rejection, and hesitation of the player $p_{1}$ for the target in all aspects, and it can fully characterise the fuzziness and uncertainty in the real game problem.

Similarly, the membership function and non-membership function of the player $p_{2}$ are also linear.
(16)$$\mu_{B}\left(x^{T}Fy\right)=\left\{\begin{array}{cc} 1 & \left(x^{T}Fy<\omega_{a}\right)\;\\ 1-\left(x^{T}Fy-\omega_{a}\right)/q_{a} & \left(\omega_{a}⩽x^{T}Fy<\omega_{a}+q_{a}\right)\;\\ 0 & \left(x^{T}Fy⩾\omega_{a}+q_{a}\right) \end{array}\right.$$
(17)$$v_{B}\left(x^{T}Fy\right)=\left\{\begin{array}{cc} 0 & \left(x^{T}Fy<\omega_{r}\right)\;\\ \left(x^{T}Fy-\omega_{r}\right)/q_{r} & \left(\omega_{r}⩽x^{T}Fy<\omega_{r}+q_{r}\right)\;\\ 1 & \left(x^{T}Fy⩾\omega_{r}+q_{r}\right) \end{array}\right.$$

A schematic diagram of the membership function and non-membership function of the player $p_{2}$ is shown in Figure 3. According to the image, it can also be interpreted similarly to the player $p_{1}$.

In the two-person zero-sum game, a gain for one player means a loss for the other. For the players $p_{1}$ and $p_{2}$, both sides want to maximise their own benefits and minimise their own losses. Since the membership function and non-membership function of player $p_{1}$ are both linear functions, the maximum-minimum strategy $x^{*}$ of player $p_{1}$ can be obtained by solving the following linear programming problem.
(18)$$\begin{array}{c} \max\{\alpha -\beta \}\;\\ \;s.t.\;\left\{\begin{array}{c} \sum_{i=1}^{m}a_{ij}x_{i}+p_{a}-v_{a}⩾p_{a}\alpha \;\left(j=1,2,\cdots ,n\right)\;\\ \sum_{i=1}^{m}a_{ij}x_{i}-v_{r}⩾-p_{r}\beta \;\left(j=1,2,\cdots ,n\right)\;\\ x_{1}+x_{2}+L+x_{m}=1\;\\ x_{i}⩾0\left(i=1,2,\cdots ,m\right)\;\\ 0⩽\alpha ⩽1,0⩽\beta ⩽1\;\\ \alpha +\beta ⩽1 \end{array}\right. \end{array}$$
Among them, α, β, $x_{i}\;\left(i=1,2,\cdots ,m\right)$ are decision variables.

Similarly, the membership function and non-membership function of player $p_{2}$ are also linear functions, thus the minimum-maximum strategy $y^{*}$ of player $p_{2}$ can be obtained by solving the following linear programming problem.
(19)$$\begin{array}{c} \min\{\eta -\lambda \}\;\\ \;s.t.\;\left\{\begin{array}{c} \sum_{j=1}^{n}a_{ij}y_{j}-\omega_{a}-q_{a}⩽-q_{a}\lambda \;\left(i=1,2,\cdots ,m\right)\;\\ \sum_{j=1}^{n}a_{ij}y_{j}-\omega_{r}⩽q_{r}\eta \;\left(i=1,2,\cdots ,m\right)\;\\ y_{1}+y_{2}+L+y_{n}=1\;\\ y_{j}⩾0\left(j=1,2,\cdots ,n\right)\;\\ 0⩽\lambda ⩽1,0⩽\eta ⩽1\;\\ \lambda +\eta ⩽1 \end{array}\right. \end{array}$$
Among them, η, λ, $y_{i}\;\left(i=1,2,\cdots ,n\right)$ are decision variables.

By solving the above two linear programming problems, the mixed strategy Nash equilibrium of the game model can be obtained.

## 5. Results

In this paper, the experiments use the following environment configuration: GPU model NVIDIA GeForce RTX 3060, 2.3 GHz CPU and 64 GB RAM.

### 5.1. Infrastructure Network Model Example

In this paper, the target infrastructure network shown in Figure 4 is used. The target network contains 16 nodes and 27 edges, of which the maximum degree of node 6 is 7, and the minimum degree of nodes 8, 12, 15, and 16 is 1. This target infrastructure network is a model of real-world infrastructure network topologies such high-speed rail networks, airport networks, and power transmission networks. Using the high-speed railway network as an example, the nodes in the complex network topology diagram represent different stations in the high-speed railway network, and the connecting edges between the nodes indicate the lines that trains travel between. In the actual world, if a station is attacked and compromised, and its train transfer function is lost, all lines that pass through the station are disabled.

In this chapter, we use the maximum connected patch size as the network performance evaluation function Γ, and we stipulate that the resources available to the attacker and the defender are equal, both of which are 4—that is, $n_{A}=n_{D}=4$.

### 5.2. Results of Single Attack and Defense Game

The experimental steps of this section are as follows:Generate adjacency matrix. According to the topological structure diagram of the target network, the corresponding adjacency matrix $A\left(G\right)={\left(a_{ij}\right)}_{N\times N}$ is calculated. If there is an edge connection between the node $v_{i}$ and the node $v_{j}$, then $a_{ij}=a_{ji}=1$, otherwise, $a_{ij}=a_{ji}=0$.Generate a revenue matrix. The attacker gives an attack strategy, and the defender gives a defence strategy. Such a combination of attack and defence strategies constitutes a strategy profile. By traversing all strategy combinations, the gains of attackers and defenders under different strategy profiles are calculated, and finally the gain matrix is obtained.Solve the mixed strategy Nash equilibrium. Based on the solution method in the previous section, we set the attacker’s expectation value $v_{a}$ for the target, and the corresponding error $p_{a}$; furthermore, the attacker’s rejection value of the target $v_{r}$ corresponds to the error $p_{r}$. Set the defender’s expectations of the target value $w_{a}$ and the corresponding error $q_{a}$; moreover, the defender’s rejection of the target value $w_{r}$ correspond to the error $q_{r}$. By calling the MATLAB linear programming solver, the mixed strategy Nash equilibrium is obtained.

In this experiment, we stipulated that the resources available to the attacker and the defender are equal, that is $n_{A}=n_{D}=4$, which means the attacker can only attack 4 nodes at a time, and the defender can only defend 4 nodes at a time. Thus, the attacker’s policy set contains a total of $C_{16}^{4}=1820$ policies; similarly, the defender’s strategy set contains a total of $C_{16}^{4}=1820$ strategies, and the number of strategy profiles reaches 3, 312,400. By traversing these policy profiles, the payoff function is used to calculate the payoff of attackers and defenders under different policy profiles, and the corresponding payoff matrix is obtained.

According to the payoff matrix, we set the attacker’s expectation of the target $v_{a}=0.4$, which corresponds to the error $p_{a}=0.3$; furthermore, the attacker’s rejection value of the target is $v_{r}=0.25$, which corresponds to the error $p_{r}=0.2$. That is, when the attack strategy reduces the performance of the target network by less than 5%, the attacker’s satisfaction degree is 0, and the rejection degree is 1; when the attack strategy reduces the performance of the target network by 5∼10%, the attacker’s satisfaction is 0, but because the attacker has a certain degree of hesitation, it will not be completely rejected at this time; when the attack strategy reduces the performance of the target network by 25∼40%, the attacker’s rejection degree is reduced to 0, but because the attacker has a certain degree of hesitation, it is not completely satisfied at this time; and when the attack strategy reduces the performance of the target network by more than 40%, the attacker’s satisfaction degree is 1 and the rejection degree is 0. The image of the attacker ’s membership and non-membership functions is shown in Figure 5.

Set the defender’s expectations of the target value $w_{a}=0.1$, and the corresponding error is $q_{a}=0.2$; the defender’s rejection of the target value is $w_{r}=0.15$, which corresponds to the error $q_{r}=0.25$. The specific explanation is similar to the attacker. The membership function and non-membership function images of the defender are shown in Figure 6.

The MATLAB linear programming solver was called, and the results of the mixed strategy Nash equilibrium are shown in Table 2. The pure strategy with a probability that was not equal to 0 in an equilibrium state and its corresponding equilibrium probability are given in the table. It can be seen from the equilibrium results that the attacker had 11 pure strategies with a probability greater than 0 to choose from, among which the probability assigned to { 3, 6, 13, and 14 } was the highest, and the degree of the remaining nodes was not prominent except for 6 nodes. The defender had 10 pure strategies with a probability greater than 0 to choose from, among which the probability assigned to { 1, 3, 9, and 11 } was the highest, and the degree of these four nodes was more prominent.

When the attacker chooses the hybrid Nash equilibrium strategy, the minimum degree of reaching the expected goal is 0.5268, the maximum degree of reaching the rejection goal is 0, and the degree of hesitation is 0.4732. When the defender chooses the mixed strategy Nash equilibrium, the minimum degree of reaching the expected goal is 0.4322, the maximum degree of reaching the rejection goal is 0.2098, and the degree of hesitation is 0.3580.

In order to further explore the preferences of attackers and defenders to attack and defend nodes, we mapped the mixed strategy probability distribution to nodes as shown in Equations (19) and (20) as follows:(20)$$\rho_{A}=\frac{1}{n_{A}}\sum_{i=1}^{\left|S_{A}\right|}p_{i}\cdot X_{i}=\frac{1}{n_{A}}\sigma_{A}\cdot X_{\left|S_{A}\right|\times N}$$
(21)$$\rho_{D}=\frac{1}{n_{D}}\sum_{j=1}^{\left|S_{D}\right|}q_{j}\cdot Y_{j}=\frac{1}{n_{D}}\sigma_{D}\cdot Y_{\left|S_{D}\right|\times N}$$

Among them, $\rho_{A}=\left[{\tilde{p}}_{1},{\tilde{p}}_{2},\ldots ,{\tilde{p}}_{i},\ldots ,{\tilde{p}}_{N}\right]$ represents the probability that the attacker allocates to each node in the Nash equilibrium, and $\rho_{D}=\left[{\tilde{q}}_{1},{\tilde{q}}_{2},\ldots ,{\tilde{q}}_{j},\ldots ,{\tilde{q}}_{N}\right]$ represents the probability the defender allocates to each node in the Nash equilibrium. $\sigma_{A}=\left[p_{1},p_{2},\ldots ,p_{i},\ldots ,p_{\left|S_{A}\right|}\right]$ and $\sigma_{D}=\left[q_{1},q_{2},\ldots ,q_{j},\ldots ,q_{\left|s_{D}\right|}\right]$ denote the probability distribution of all pure strategies in the Nash equilibrium.

The probability distribution of the equilibrium results mapped to different nodes in Table 2 is shown in Table 3. From Table 3, it can be seen that attackers pay high attention to nodes 1, 3, 6, 12, 13, and 14, while the degree and betweenness of nodes 1, 3, 13, and 14 are at a medium level. In addition, nodes 12, and even edge nodes, indicated that, when attacking the network, the attackers rarely attacked nodes with high importance in the network, but rather paid more attention to the nodes at a medium level and betweenness in the network. The attackers improved their success rate and the revenue of their attacks through such attack strategies. On the other hand, the defenders had the highest probability of defending node 11, which had the largest betweenness and highest degree in the network, thereby indicating that defenders are more inclined to protect the more important nodes in the network to ensure the normal performance of the network. In addition, the betweenness of nodes 2, 8, 15, and 16 was 0, and the probability that the attacker and the defender chose the node was also 0, thus indicating that these nodes were not of concern by both sides.

The probability diagrams of different nodes being attacked or defended in the equilibrium state are respectively shown in Figure 7.

In order to verify the effectiveness of the game model in obtaining the equilibrium strategy, we selected the maximum degree strategy, the minimum degree strategy, and the random strategy to compare with the equilibrium strategy. The maximum degree strategy refers to the attack and defence of the largest nodes in the network. In contrast, if the offensive and defensive sides attack and defend the smallest nodes in the network, then this represents the minimum degree strategy; the random strategy means that the attacker and the defender do not consider the indicators of the nodes and randomly select several nodes in the network to attack and defend. Table 4 shows the gains of attackers when attackers and defenders adopt four different respective strategies. Due to the great uncertainty of the selection of random strategies, we conducted 3000 independent experiments and used the average of the final results as the final equilibrium result.

We calculated the Nash equilibrium again for the equilibrium income matrix that was obtained in Table 3, and the results were ( [ 1, 0, 0, 0 ], [ 1, 0, 0, 0 ] ), that is, (equilibrium strategy, equilibrium strategy) was the Nash equilibrium point. Further, both the offensive and defensive sides chose the equilibrium strategy to obtain the maximum income, which further verifies the effectiveness of the equilibrium strategy that was obtained in this section.

### 5.3. Results of Repeated Network Attack and Defense Game

In the previous section, we only studied the situation of one round in a game. In reality, the offensive and defensive sides will carry out multiple rounds of attack and defence around complex networks such as traffic networks and communication networks. The end state of each round of game will become the starting state of the next round of the game. Therefore, in order to be closer to the reality, we will repeat the game around the target network in this section.

In the repeated game model, we will add the following assumptions: (1) At the beginning of each round of a game, the strategy choice of the attacker and the defender is only related to the topology of the complex network in the current state, and it is not affected by the strategy choice of the two sides in the previous round of the game—that is, the repeated game has a Markov property. (2) The starting state of each round of the game is the current complex network model, and the ending state is the complex network model after deleting the successfully attacked nodes. Moreover, the ending state of that round of the game is the starting state of the next round of the game. (3) The termination condition of the repeated game is as follows: no matter what attack strategy the attacker adopts, the repeated game terminates when the node in the target network cannot be successfully attacked.

The remaining assumptions are the same.

Based on the above assumptions, the design of repeated game experiment steps was as follows:Calculate the adjacency matrix.Generate the income matrix.Solve for the mixed strategy Nash equilibrium based on the equilibrium payoff matrix.Solving for mixed strategy Nash equilibrium based on the equilibrium payoff matrix.Based on the mixed-strategy Nash equilibrium result, the attacker and the defender each select a pure strategy to play. We adopted a “roulette” approach for selecting pure strategies from among the equilibrium strategies.Obtain the number of nodes that were successfully attacked and calculate the value of the attacker’s gain in this game. The defender’s gain is the negative of the attacker’s gain.Update the complex network diagrams.

After modelling and solving, this repeated game was played 5 times to reach the termination state, and the progress of the repeated game is shown in Figure 8. For ease of record keeping, we used Max, Min, Random and Nash to represent the maximum degree strategy, minimum degree strategy, random strategy, and equilibrium strategy, respectively. The attacker’s strategy choice for each round of the game was Nash → Min → Nash → Nash → Nash; the defender’s strategy choice for each round of the game was Nash → Nash → Nash → Nash → Nash.

In the repeated game process, we found that the attacker chose the minimum degree strategy in the second round of the game, and that the gain was higher than that of the equilibrium strategy. That is to say, although the equilibrium strategy will bring more robust benefits, the equilibrium strategy is not the optimal choice in certain cases. For the defender, he has always chosen the equilibrium strategy in the repeated game process, thereby indicating that, although other strategies may bring higher earnings, the defender cannot bear the risks brought by other strategies.

The actual income in each round of the game and the equilibrium income in the current state are given in Figure 9. We find that the actual earnings of each round of the game was not the same as the equilibrium earnings in the current state. This is because we used the ’roulette’ method to select the pure strategy from the equilibrium strategy, which results in a great deal of randomness.

Now, we assume that, if the attacker knows the defender’s defence strategy before choosing the attack strategy every time, then the attacker will perfectly avoid the defender’s choice of the defender’s node every time they choose the attack node. Under this assumption, the attacker only needs to carry out three rounds of attacks to achieve the termination condition of the repeated game. However, in the experiment of this section, the attacker carried out a total of five rounds of attacks to terminate the repeated game. The target network used in this paper only contains 16 nodes. However, the scale of complex networks in reality is huge. Repeated games on such networks may take thousands of rounds to reach the termination condition. It is of great practical significance to study the convergence of this problem and how to reach the termination condition of repeated games more quickly.

## 6. Discussion

It is difficult for players to accurately assess the game aim in the game process due to the availability of complicated components such as information, knowledge, and subjective consciousness of the players involved in the real game. As a result, the player’s impression of the game goal is ambiguous. It is precisely because of this uncertainty that the players would be hesitant about the degree of achievement of the game aim. The intuitionistic fuzzy set can be used to represent the degree to which the player accomplishes the predetermined goal and the degree of hesitancy to achieve the predetermined goal while picking the strategy, which gives the players more information to choose their strategy.

The model established by this method not only considers objective conditions such as network topology and payoff function when solving the equilibrium strategy, but it also quantitatively analyzes and evaluates subjective factors such as decision-makers’ preferences and hesitation. Simultaneously, this paper for the first time introduces the two-player zero-sum game model with the goal of intuitionistic fuzzy set into the complex network attack and defense game, broadening the boundary of research on the attack and defense game of infrastructure network and providing a new research idea for solving the uncertainty problem of quantitative analysis in the field of attack and defense game of infrastructure network. Therefore, the model established in this paper has good theoretical and practical significance.

However, in the course of creating the model, this study adds additional model assumptions and strongly limits the model’s field of use. In reality, obtaining the complete information required in this paper is impossible, and typically, the defender has a complete understanding of the network topology information, whereas the attacker has only limited information, which also contains false information intentionally created by the defender. Furthermore, because the importance and scale of nodes in the critical infrastructure network vary, the defender’s investment cost for each node varies, as does the attacker’s cost of assaulting each node. Furthermore, both defense and attack have a success rate, which ignores the reality that the node can still perform part of its duties after being attacked. Finally, this model does not work with several participants. This will become new study directions in the field of infrastructure network attack and defense games.

## 7. Conclusions

In this paper, a two-person, zero-sum game model, with the goal of intuitionistic fuzzy sets, was introduced into the research on complex network attack–defence games. In order to simplify the problem, we make basic assumptions in the model, as well as conduct modelling and research on the basis of assumptions. We found that, when attacking the network, attackers rarely attack the nodes with high importance in the network; instead, they pay more attention to the nodes with moderate importance and betweenness in the network, whereas defenders are more inclined to protect the more important nodes in the network to ensure the normal performance of the network. After this finding, we compared the obtained equilibrium strategy with three typical strategies, and we found that both offensive and defensive sides chose the equilibrium strategy to obtain the maximum benefit, which further verifies the effectiveness of the equilibrium strategy obtained in this paper. Finally, we conducted a repeated game modelling to simulate the actual situation in reality, and we found that, for attackers (although the equilibrium strategy will bring more robust benefits), the equilibrium strategy was not the optimal choice in certain cases; for defenders, although other strategies may bring higher earnings, the defender could not bear the risks brought by other strategies. The game model in this paper provides a new research framework and ideas for later network attack and defence game modelling. The paper concludes by discussing the theoretical and practical significance of the proposed model, as well as its application scope.

## Figures and Tables

**Figure 1 entropy-25-01558-f001:**
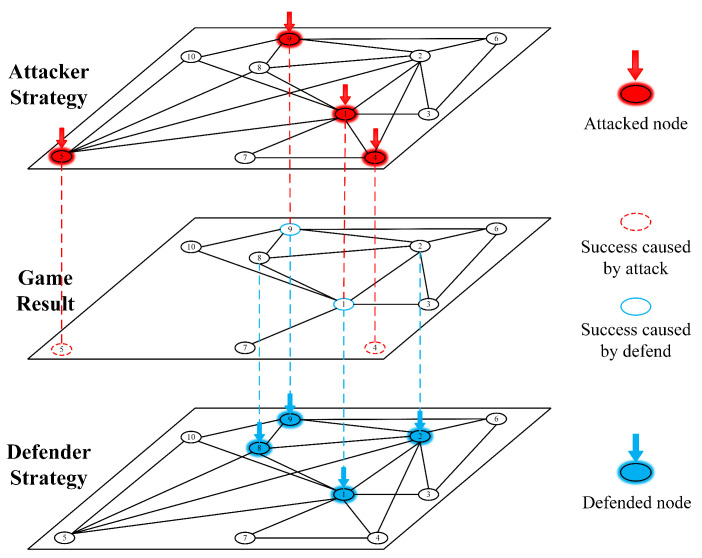
This figure shows the process of attack and defense game for infrastructure networks. The nodes in this picture that are covered in blue are those that the defender chooses to defend, and the nodes that are covered in red are those that the attacker decides to attack. The nodes with the red dotted line indicate that the attacker has successfully attacked them when it is not defended, and the nodes with the blue solid line indicate that the defender has successfully defended them.

**Figure 2 entropy-25-01558-f002:**
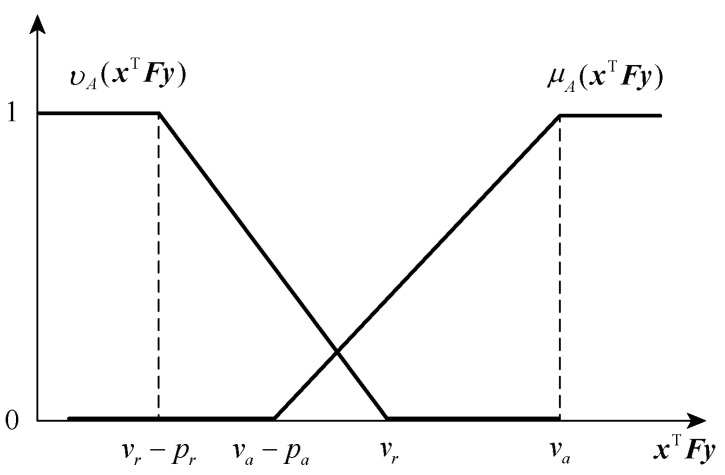
This figure shows the membership function and non-membership function image of player $p_{1}$. And the membership function monotonically non-decreasing as the payment value increases; While the non-membership function monotonically non-increasing as the payment value increases.

**Figure 3 entropy-25-01558-f003:**
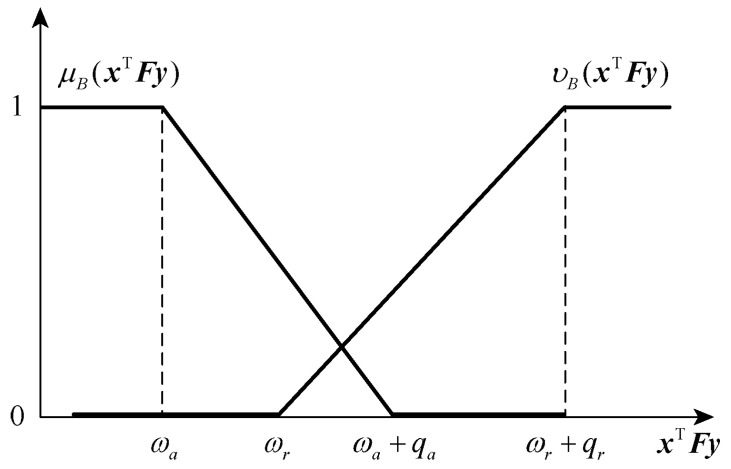
This figure shows the membership function and non-membership function image of player $p_{2}$. And the membership function monotonically non-increasing as the payment value increases; While the non-membership function monotonically non-decreasing as the payment value increases.

**Figure 4 entropy-25-01558-f004:**
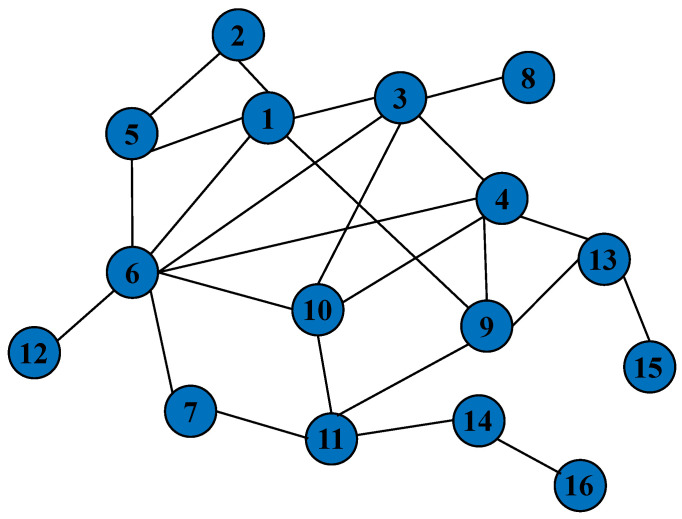
This figure shows the topology of the target infrastructure network. Nodes represent stations with specialized purposes in real-world complex networks, such as high-speed train stations, airports, and power stations; A linked edge is defined as a physical or logical connection between two nodes that allows them to exchange information or matter.

**Figure 5 entropy-25-01558-f005:**
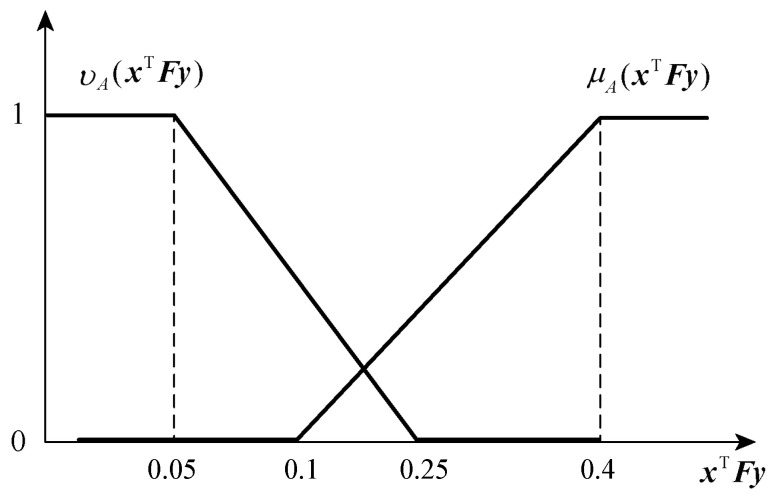
This figure shows the image of the attacker’s membership function and non-membership function. And the image of attacker’s membership and non-membership function are obtained with $v_{a}=0.4$, $p_{a}=0.3$, $v_{r}=0.25$, $p_{r}=0.2$.

**Figure 6 entropy-25-01558-f006:**
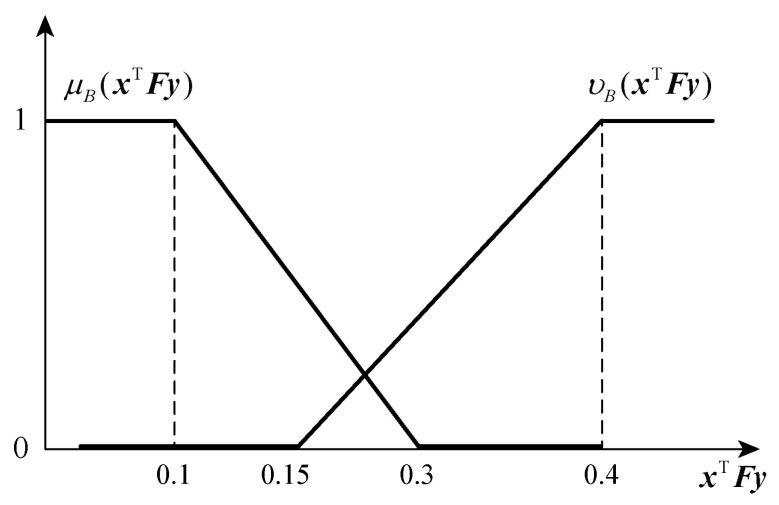
This figure shows the image of the defender’s membership function and non-membership function. And the image of defender’s membership and non-membership function are obtained with $w_{a}=0.1$, $q_{a}=0.2$, $w_{r}=0.15$, $q_{r}=0.25$.

**Figure 7 entropy-25-01558-f007:**
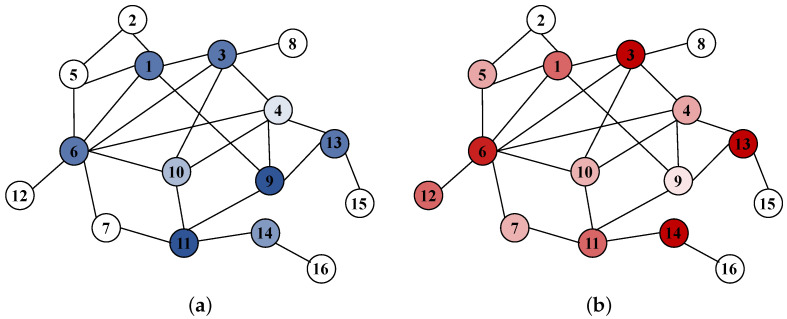
This picture shows the probability of different nodes being attacked or defended. The darker the colour is, the greater the probability that the node is attacked or defended. And a node with no color indicates that it is not selected by attackers or defenders. (**a**) The probability of different nodes being defended. (**b**) The probability of different nodes being attacked.

**Figure 8 entropy-25-01558-f008:**
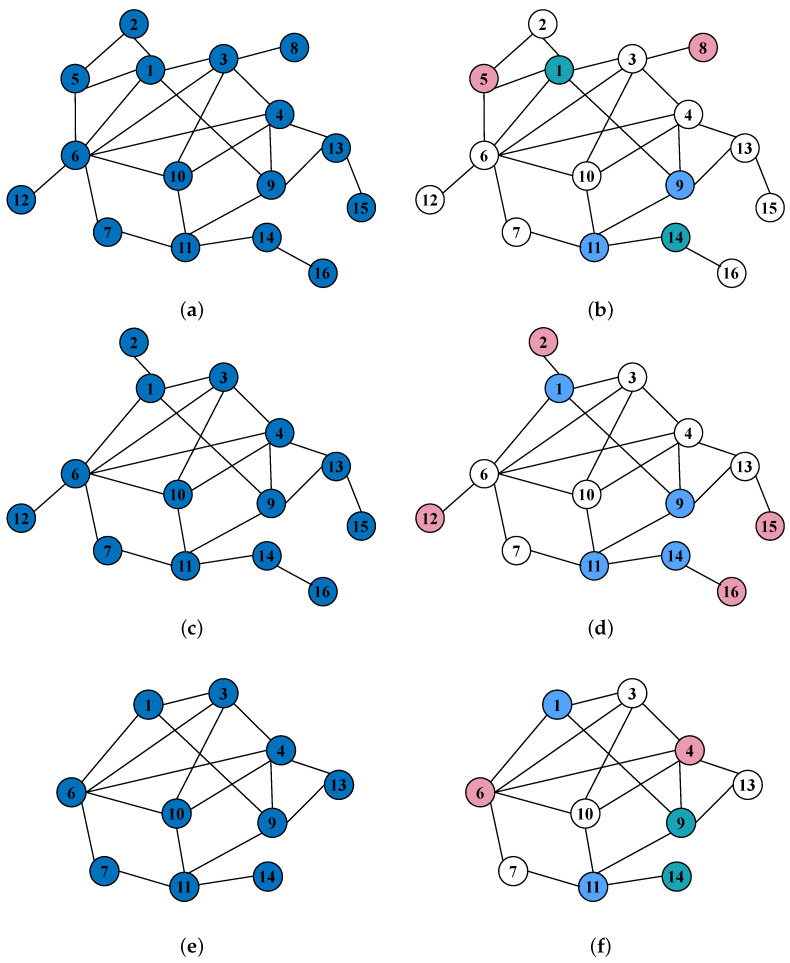
This picture shows the specific process of each round of a game in the process of a repeated game. Among them, the picture of the left column represents the topology of the infrastructure network at the beginning of each round of the attack and defence game. The picture in the right column shows the strategy choice of each round of attackers and defenders. The marked red node represents the node that successfully attacked, the green node represents the node that successfully defended, and the blue node represents the node that was defended but not attacked. (**a**) Topology of the first round of the complex network. (**b**) The first round of attack and defence strategy. (**c**) Topology of the second round of the complex network. (**d**) The second round of attack and defence strategy. (**e**) Topology of the third round of the complex network. (**f**) The third round of attack and defence strategy. (**g**) Topology of the fourth round of the complex network. (**h**) The fourth round of attack and defence strategy. (**i**) Topology of the fifth round of the complex network. (**j**) The fifth round of attack and defence strategy.

**Figure 9 entropy-25-01558-f009:**
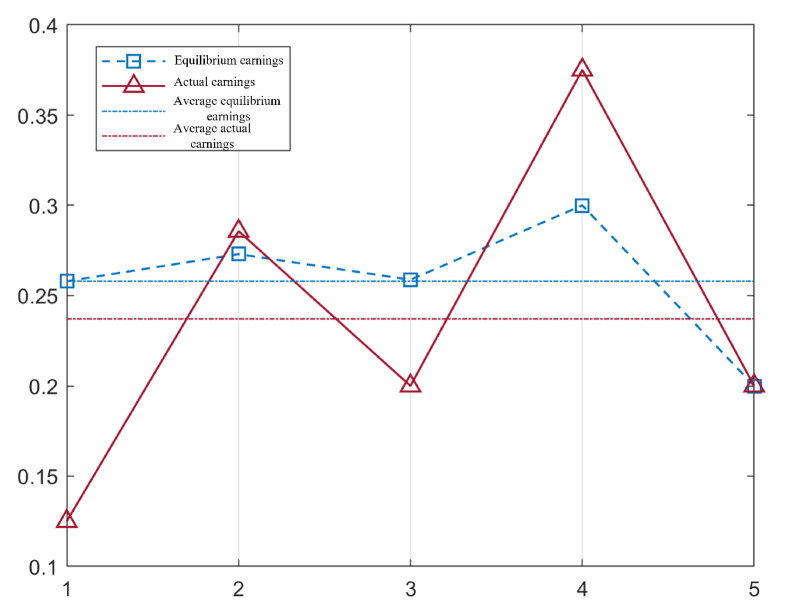
This figure shows the comparison between the actual earnings and the equilibrium earnings under the repeated game. The theoretical equilibrium and actual earnings of each game are shown by the blue box and red triangle nodes. And the blue dotted and red dotted lines represent the average equilibrium earnings and the average actual earnings.

**Table 1 entropy-25-01558-t001:** Summary of the application of game theory in the field of attack and defence game for infrastructure networks.

Game Model	Author	Research Methods
Starkberg game model	Zeng [25,26]	Generate a pseudo network, disclose false network structure information to attackers, and use asymmetric information to actively defend against cost constraints.
	Gu [23]	Analyze the influence of different utility function types of attackers on the equilibrium solution.
	Jiang [24]	Developed a Bayesian Starkelberg game model to study the water supply network protection problem, including four private information situations.
Dynamic game model	Borwn [17]	Establish two-layer and three-layer planning models.
	Fu [22]	The defender first protects the network through protection or camouflage behaviors, then the attacker takes measures to destroy it.
	Wang [33]	The selection of active defense strategies is carried out in the form of pure policies.
Two-player zero-sum game model	Li [19,20,21]	Define the strategy and benefit according to the topology of the infrastructure system, and construct the benefit matrix by enumerating all the strategies to solve the problem.
Defender-attacker-defender sequential game model	Jiang [34], Thompson [27,28], Yuan [29]	The offensive and defensive process is modeled into three phases: (1) the defender invests in the infrastructure system with limited resources; (2) the attacker sees these investments and attacks the system to maximize the damage effect; (3) The defender repairs the network at the minimum cost according to the existing investment and attack to reduce losses, such as swapping edges and reducing load.
Asynchronous game model	Zhang [8]	Defenders allocate defensive resources to construct false targets to fortify real targets; Attackers allocate resources to identify false targets and attack real targets, enabling the allocation of defense resources among multiple targets.

**Table 2 entropy-25-01558-t002:** Mixed strategy probability distribution table under equilibrium state.

Attack Strategy	Probability	Defence Strategy	Probability
{ 1, 3, 5, 14 }	0.0104	{1, 3, 6, 9 }	0.1077
{ 1, 4, 6, 11 }	0.0414	**{ 1, 3, 9, 11 }**	**0.2500**
{ 1, 4, 7, 10 }	0.0473	{ 1, 6, 9, 13 }	0.0828
{ 1, 5, 6, 11 }	0.0518	{ 1, 6, 13, 14 }	0.0594
{ 1, 5, 12, 14 }	0.1760	{ 3, 4, 6, 14 }	0.0947
{ 1, 6, 13, 14 }	0.0207	{ 3, 10, 11, 14 }	0.0476
{ 3, 5, 6, 11 }	0.0104	{ 6, 9, 13, 14 }	0.0361
{ 3, 6, 11, 13 }	0.0207	{ 6, 10, 11, 13 }	0.1238
**{ 3, 6, 13, 14 }**	**0.2899**	{ 9, 11, 13, 14 }	0.1297
{ 3, 11, 12, 13 }	0.1657	{ 10, 11, 13, 14 }	0.0727
{ 4, 7, 9, 10 }	0.1657		

**Table 3 entropy-25-01558-t003:** Probability distribution tables on different nodes.

Node	1	2	3	4	5	6	7	8
Degree	5	2	5	5	3	7	2	1
Betweenness	16.9	0	14.5	14.5	2.1	**29.5**	4	0
Attack probability	0.087	0	**0.124**	0.064	0.062	**0.109**	0.053	0
Defence probability	**0.125**	0	**0.125**	0.024	0	**0.125**	0	0
Node	9	10	11	12	13	14	15	16
Degree	4	4	4	1	3	2	1	1
Betweenness	**21.1**	11.5	**29**	0	14	14	0	0
Attack probability	0.041	0.053	0.073	0.085	**0.124**	**0.124**	0	0
Defence probability	**0.151**	0.061	**0.156**	0	**0.125**	0.109	0	0

**Table 4 entropy-25-01558-t004:** Table of equilibrium gains for the attacker under four different strategies.

Attack	Defense			
	Nash	Max	Min	Random
**Nash **	**0.2580**	0.2836	0.5072	0.3541
Max	0.2441	0	0.5	0.3495
Min	0.25	0.25	0	0.1871
Random	0.2266	0.2358	0.2939	0.2562

## Data Availability

The data presented in this study are available on request from the corresponding author. The data are not publicly available due to some privacy reasons.

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
