# Peer review of "Modelling and Research on Intuitionistic Fuzzy Goal-Based Attack and Defence Game for Infrastructure Networks"

_entropy, 2023, doi:10.3390/e25111558_

Round 1

Reviewer 1 Report

Comments and Suggestions for Authors

In "Modelling and Research on Intuitionistic Fuzzy Goal-Based Attack and Defence Game for Infrastructure Networks" the authors study an interesting problem and present new results related to network resilience against attacks and how this might be achieved in the context of fuzzy goal-based attacks on infrastructure networks.

I have enjoyed reading, and I have only a few comments that require revision. If this is carried out, I would expect the manuscript to be suitable for publication in Entropy.

Research points out the ever-growing importance of network resilience against targeted attack and error. This has been emphasized recently in the reviews Signal propagation in complex networks, Peng Ji, et al., Phys. Rep. 1017, 1-96 (2023) and in the research paper Evolution of cooperation on scale-free networks subject to error and attack, M. Perc, New J. Phys. 11, 033027 (2009), where attack and games on a network have been combined for the first time. This should further improve the introduction and the outline of possible future research.

It would also improve the paper if the figure captions would be made more self-contained. In addition to what is shown for which parameter values, one could also consider a sentence or two saying what is the main message of each figure.

Some references contain errors, missing or incorrect information, and inconsistent formatting. It is difficult to give credit to research if such elementary aspects of the work are not error free. References should thus be corrected with the best care.

Finally, the authors should please share their source code in a repository. This would promote the usage of the proposed model and allow also others to take better advantage of this research, and also to allow them to reproduce the results.

If a revision is granted, I will be happy to review the manuscript again.

Reviewer 2 Report

Comments and Suggestions for Authors

This paper introduces a new approach to modeling offensive and defensive games from a network science perspective. This paper considers the attacker’s and defender’s expectation values, rejection values, and hesitation degrees regarding the target to construct an intuitionistic fuzzy goal-based attack and defense game model for infrastructure networks. The new approach proposed in this paper has potential applications in protecting infrastructure networks against attacks by terrorist organizations. While the manuscript showcases several strengths, there are some issues that must be addressed before it can be considered for publication.

1. The introduction lacks a clear presentation of the contributions of this article. It is necessary to explicitly state the strengths and innovative aspects of the paper.

2. In section 3.4, the network performance evaluation function lacks a clear and specific definition.

3. In equation 4, the element "F" in the multivariate array lacks explanatory descriptions, making it difficult for readers to understand how to obtain an intuitionistic fuzzy set.

4. In line 202, the research by Nan and Li requires a citation.

5. The results lack a detailed explanation of the experimental environment and data. In the infrastructure network model, it is unclear what specific facilities different nodes represent, how node states are defined, and the exact nature of the attack and defense strategies. Providing concrete examples for clarification would be beneficial.

6. In Figure 9, there are duplicate subgraph labels.

7. There are some grammatical issues in the paper, such as the dash in line 160.

Comments on the Quality of English Language

There are some grammatical issues in the paper that require moderate editing of the English language.

Reviewer 3 Report

Comments and Suggestions for Authors

In this paper, the authors present a new intuitionistic fuzzy goal-based game model to address the lack of consideration for uncertainty factors—such as the subjective preferences of attackers and defenders—in the study of infrastructure network protection. The model incorporates the psychological aspects of expectation, rejection, and hesitation and translates the network defense game into a linear programming problem. Through analysis and experiments in various game scenarios, the findings reveal that attackers preferentially target less critical nodes, contrary to the defenders' strategy of protecting the more vital ones, thus ensuring the network's operational integrity.

The paper is well written and flows well. The theoretical foundations are solid and well defined. The whole manuscript was a pleasure to read and I found it very interesting. Nevertheless, there are some aspects that the authors should improve to improve the quality of the manuscript:

  • I appreciate the effort the authors made in presenting the related studies overview. However, it is not easy to understand how the current approach is positioned w.r.t. the mentioned studies. Therefore, starting from the previous works, I suggest introducing a table to summarize the most recent works and to highlight the novelty of the proposed work.
  • Along the same lines of the previous point, the authors could consider citing the work in [1]. In fact, the method therein proposed could be used to define a similarity measure between the mixed strategies which could account for heterogeneity and uncertainty.
  • Although the method is described soundly and the experimental evaluation is solid, the implications are not very clear. I suggest the authors to provide a discussion about the theoretical and practical implications of such work.
  • The authors could discuss the potential limitations of the approach. For instance, could the model assumptions (Sect. 3.1) pose too strict limitations? What are the edge cases in which the proposed approach could fail?
  • A table of the used variables should be reported.
  • Line 168: loosely speaking, X is a one-hot encoding of the strategy. Is that correct? If so, the authors should highlight it.
  • Line 171: remove the comma between the sum and x_i.
  • Line 180: define \hat{E}
  • Explain the ranges in Equations 2 and 3.
  • Line 202: insert the correct reference to Nan and Li.

References:

[1] Generalizing identity-based string comparison metrics: Framework and techniques. Knowledge-Based Systems, 187, 104820.

Round 2

Reviewer 2 Report

Comments and Suggestions for Authors

The author's responses to my concerns are satisfactory. I recommend accepting the manuscript for publication in its current form.

Reviewer 3 Report

Comments and Suggestions for Authors

The authors successfully addressed my concerns. In my opinion, the manuscript can be accepted for publication.